# A model of tuberculosis clustering in low incidence countries reveals more transmission in the United Kingdom than the Netherlands between 2010 and 2015

Ellen Brooks-Pollock[1,2]*, Leon Danon[3,4], Hester Korthals Altes[5], Jennifer A. Davidson[6], Andrew M. T. Pollock[7], Dick van Soolingen[5,8], Colin Campbell[6], Maeve K. Lalor[6,9]

**1** Population Health Sciences, Bristol Medical School, University of Bristol, Bristol, United Kingdom, **2** Bristol Veterinary School, University of Bristol, Bristol, United Kingdom, **3** College of Engineering and Mathematical Sciences, University of Exeter, Exeter, United Kingdom, **4** The Alan Turing Institute, London, United Kingdom, **5** Centre for Infectious Disease Control, National Institute for Public Health and the Environment, Bilthoven, the Netherlands, **6** TB Section, Public Health England, London, United Kingdom, **7** Department of Physics, University of Sheffield, Sheffield, United Kingdom, **8** Departments of Clinical Microbiology and Pulmonary Diseases, Radboud University Nijmegen Medical Center, Nijmegen, The Netherlands, **9** Institute for Global Health, University College London, London, United Kingdom

* Ellen.Brooks-Pollock@bristol.ac.uk

## Abstract

Tuberculosis (TB) remains a public health threat in low TB incidence countries, through a combination of reactivated disease and onward transmission. Using surveillance data from the United Kingdom (UK) and the Netherlands (NL), we demonstrate a simple and predictable relationship between the probability of observing a cluster and its size (the number of cases with a single genotype). We demonstrate that the full range of observed cluster sizes can be described using a modified branching process model with the individual reproduction number following a Poisson lognormal distribution. We estimate that, on average, between 2010 and 2015, a TB case generated 0.41 (95% CrI 0.30,0.60) secondary cases in the UK, and 0.24 (0.14,0.48) secondary cases in the NL. A majority of cases did not generate any secondary cases. Recent transmission accounted for 39% (26%,60%) of UK cases and 23%(13%,37%) of NL cases. We predict that reducing UK transmission rates to those observed in the NL would result in 538(266,818) fewer cases annually in the UK. In conclusion, while TB in low incidence countries is strongly associated with reactivated infections, we demonstrate that recent transmission remains sufficient to warrant policies aimed at limiting local TB spread.

## Author summary

Multiple tuberculosis (TB) cases infected with a single strain are known as a TB cluster. In the United Kingdom (UK) for example, TB clusters vary in size from two cases up to over 200 cases. Previous work on cluster sizes demonstrated that highly infectious individuals

**Data Availability Statement:** All relevant data are within the manuscript and its Supporting Information files.

**Funding:** EBP was supported by the National Institute for Health Research Health Protection Research Unit (NIHR HPRU) in Evaluation of Interventions. The views expressed are those of the author(s) and not necessarily those of the NHS, the NIHR or the Department of Health. LD gratefully acknowledges the financial support of The Alan Turing Institute under the EPSRC grant EP/N510129/1. The funders had no role in study design, data collection and analysis, decision to publish, or preparation of the manuscript.

**Competing interests:** The authors have declared that no competing interests exist.

influence cluster size, but the analysis did not include the largest clusters. Here, we show that the chance of observing a cluster of a given size follows the same pattern in the UK and the NL. Using a new mathematical description of how clusters are formed, we are able to predict the chance of observing the full range of cluster sizes. Using the model, we estimate how many cases are due to recent transmission and how many other cases each case generates. Although we estimate that a minority of cases (39% (26%,60%) in the UK) are due to recent in-country transmission, we find that reducing the onward transmission in the UK to levels in the NL would result in 538 (266,818) fewer cases annually in the UK.

## Introduction

Tuberculosis (TB) is a chronic infectious disease and a major global public health threat. In 2017, 10.0 million people developed TB and 1.6 million people died from TB worldwide[1]. In many low TB incidence countries, a high proportion of cases occur in persons born abroad, and control measures such as migrant screening have been introduced to limit imported infection and reduce treatment costs[2,3]. However, it is often not known whether foreign-born individuals were exposed to TB before or after immigrating [4,5], which affects the impact of such interventions.

Over the past 20 years, genotyping has informed our knowledge of how TB evolved, spread around the world, and survives within hosts[6–8]. Unlike genotyping for other pathogens, TB genotyping cannot always definitively identify who-infected-whom, as epidemiologically-linked cases are often infected with genetically indistinguishable strains[9,10]. Furthermore, cases infected by indistinguishable strains may be epidemiologically unrelated, due to infection with a common strain[11]. Instead, genotyping is often used to rule out transmission, for instance between household members infected with different strains[12,13]. In low incidence countries, distinguishable strains are used to estimate the fraction of cases that are not due to recent transmission, but due instead to the reactivation of existing infections or cases infected elsewhere[14].

TB clusters are defined as multiple cases infected with a single genotype. Clusters are often assumed to signify sustained recent transmission and factors such as pulmonary disease and country-of-origin increase an individual's risk of being part of a cluster[15]. For acute infections such as measles, the observation that clusters size distributions follow a power-law has been used to indicate that the epidemiological process is at a critical point[16].

For TB, analysis of the distribution of cluster sizes has been used to estimate the genetic mutation rate in a population[17] and infer the role of super-spreading individuals[18]. The latter method was contingent on identifying transmission clusters (defined as clustered cases occurring less than two years apart): alternative methods are required to apply this method without *a priori* epidemiological knowledge of the likely index case. Furthermore, it is not known how cluster generation differs between settings with potentially differing types of migration and social contact patterns. Here, we propose and develop a method to estimate the distribution in the number of secondary cases (the reproduction number) and the percentage of cases that are due to recent (since 2010), within-country transmission from the information in cluster size distributions for TB in the UK and the NL.

## Results

In the UK, data were available for the period between 2010 and 2015, and contained 23,646 genotyped cases and 12,503 unique genotypes. 9,802 (41.5%) cases were unique genotypes and 13,844 (58.5%) cases were in clusters containing two or more individuals.

On average, 70% (42%, 98%) of clustered cases involved pulmonary disease, compared to 53% of all cases (Fig 1A). Between 2010 and 2015, 73% of cases in the UK were non-UK born. A lower proportion of those cases are diagnosed with pulmonary TB[19] (47%) compared to UK-born cases (69%). Within a cluster, 64% (10%,95%) of cases are UK-born.

In the Netherlands (NL), data were available between 2004 and 2015 and contained 8,449 genotyped cases. 3,923 (46.4%) cases were unique genotypes and 4,526 (53.6%) cases were in clusters of two or more. Limiting the analysis to cases diagnosed between 2010 and 2015 for t, there were 3,841 genotyped cases: 2,026 (52.7%) cases had a unique genotype and 1,815 (47.3%) were in clusters of two or more. 67% of all cases involved pulmonary disease, whereas within clusters, on average 70% (26%,100%) of cases were pulmonary (Fig 1B). As in the UK, non-NL born cases make up the majority, 69%. A lower proportion of non-NL born cases were diagnosed with pulmonary TB (63%) compared to NL-born cases (76%). Within a cluster, 66% (33%,100%) of cases are non-NL born.

### Modelling the distribution of cluster sizes

The distribution of cluster sizes in the UK and the NL were fitted to a power-law (KS statistics 0.013 and 0.007 respectively; p-values 0.63 and 0.07 respectively). The estimated exponent for the UK was 2.4 (2.3, 2.6) with a minimum cluster size consistent with the power-law, $x_{min}$, of 3 (1,5). The estimated exponent for the NL was higher than the UK, 2.8 (2.7, 2.9), with a minimum cluster size consistent with the power-law, $x_{min}$, of 1 (1, 1).

From the power-law model, we can predict that in the UK, 1 in 5 genotypes will occur twice or more; 1 in 160 genotypes will occur twenty times or more; and 1 in 6,000 genotypes will occur 200 times or more. In the NL, 1 in 6 genotypes will occur twice or more; 1 in 250 genotypes will occur twenty times or more; and 1 in 8,500 genotypes will occur 200 times or more.

The cluster size distribution in the NL was captured by both branching models, where the distribution of secondary cases follows either a negative binomial or a Poisson lognormal distribution (See Figs A-D in S1 Text for posterior distributions). Both models captured the number of unique genotypes and cluster sizes that occur only once (Fig 2, left, also section S1.2 in S1 Text). Although a branching process with a negative binomial distribution of secondary cases was able to capture the number of unique genotypes in the UK data, it systematically underestimated the frequency of large clusters (Fig 2, right).

A Poisson lognormal model resulted in increased model uncertainty, but provided an improved fit, and captured the entire distribution in the UK (Figs 2 and 3A) as well as still capturing the NL data (Figs 2 and 3B).

The Poisson lognormal distribution for the number of secondary cases in the UK had logmean of -2.9 (-4.7, -1.5) and log-variance 2.0 (1.2, 2.8). In NL, between 2004 and 2015 the logmean was -2.9 (-5.0, -1.6) and log-variance 1.9 (1.0, 2.8). Restricting the analysis to cases reported in NL between 2010 and 2015, decreased the log-mean to -3.4 (-6.7, -1.7) and slightly increased the log-variance 1.9 (1.0, 3.4).

### Cases due to recent transmission

From our models, we estimate that between 2010 and 2015, the percentage of cases due to recent transmission in the UK was 39% (26%,60%); with 61% (40%, 74%) of cases due to importation or reactivation. In the NL, we estimate that 23%(13%,37%) of cases are

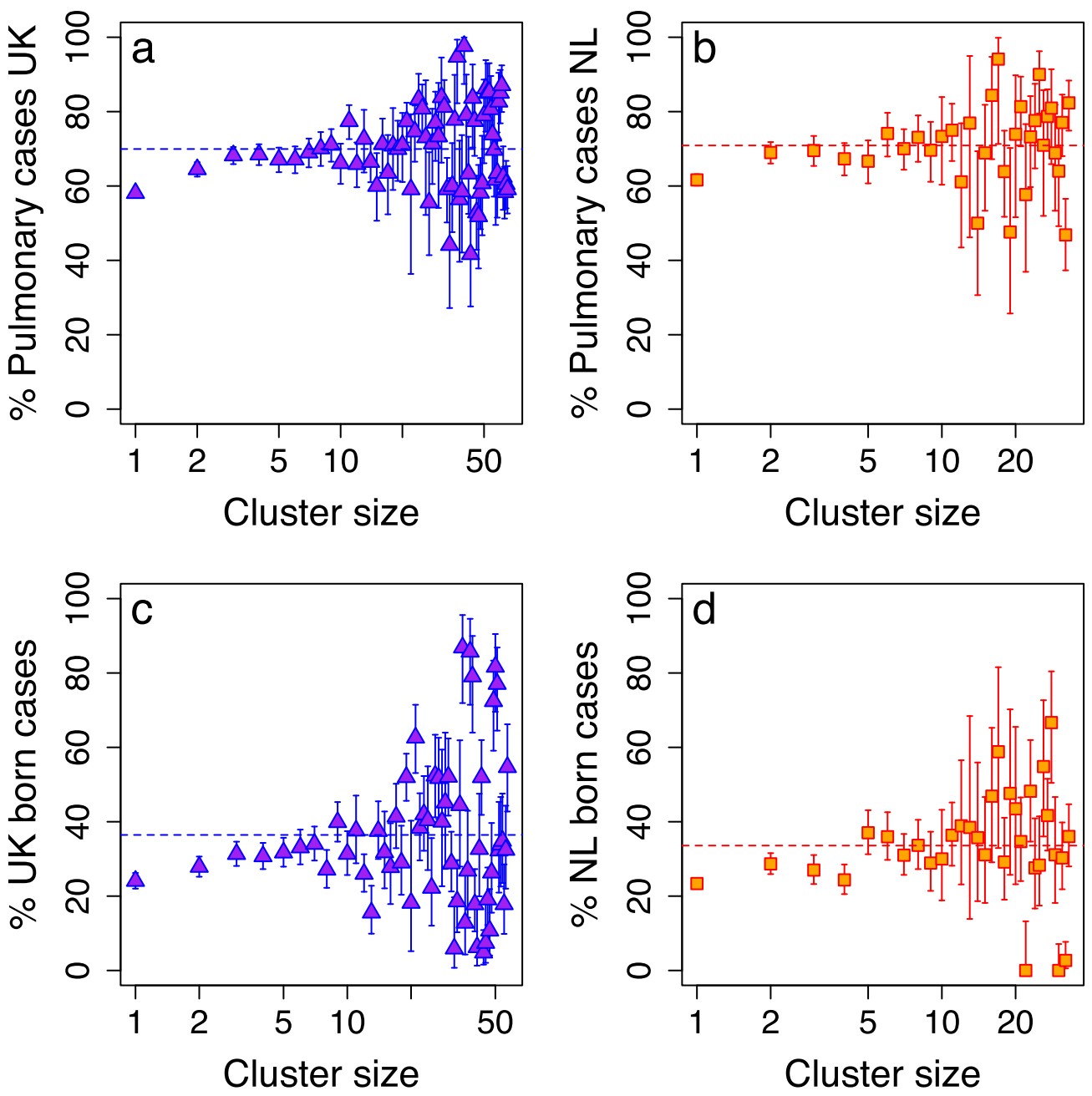

**Fig 1.** The percentage of pulmonary cases in a cluster against cluster size for the UK (a) and the Netherlands (b). The percentage of foreign-born cases in a cluster against cluster size for the UK (c) and the Netherlands (d). Dotted lines indicate the mean value for a cluster.

attributable to recent transmission, the remainder being due to importation or reactivation (see section S1.3 and Figure G in S1 Text).

### The effective reproduction number

The mean effective reproduction number is calculated from the Poisson lognormal distribution (Eq 1, Methods). For the UK, it was 0.41 (0.30, 0.60), suggesting that, on average,

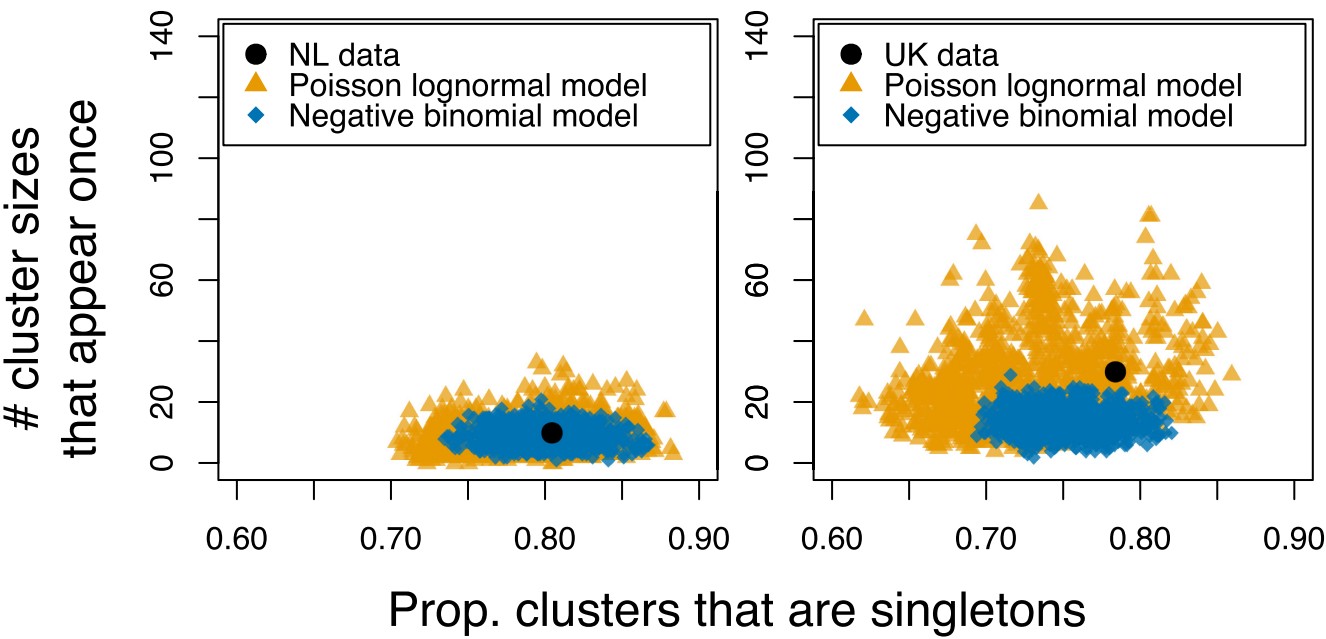

**Fig 2. The number of clusters of size 1 (i.e. unmatched cases) against the number of cluster sizes that appear exactly once.** The coloured points are 1,000 model replicates selected from the posterior distributions for the branching process model with the distribution of secondary infections following either a Poisson lognormal distribution (yellow triangles) or a negative binomial distribution (blue diamonds). The black points indicate the data values for the NL (left) and the UK (right).

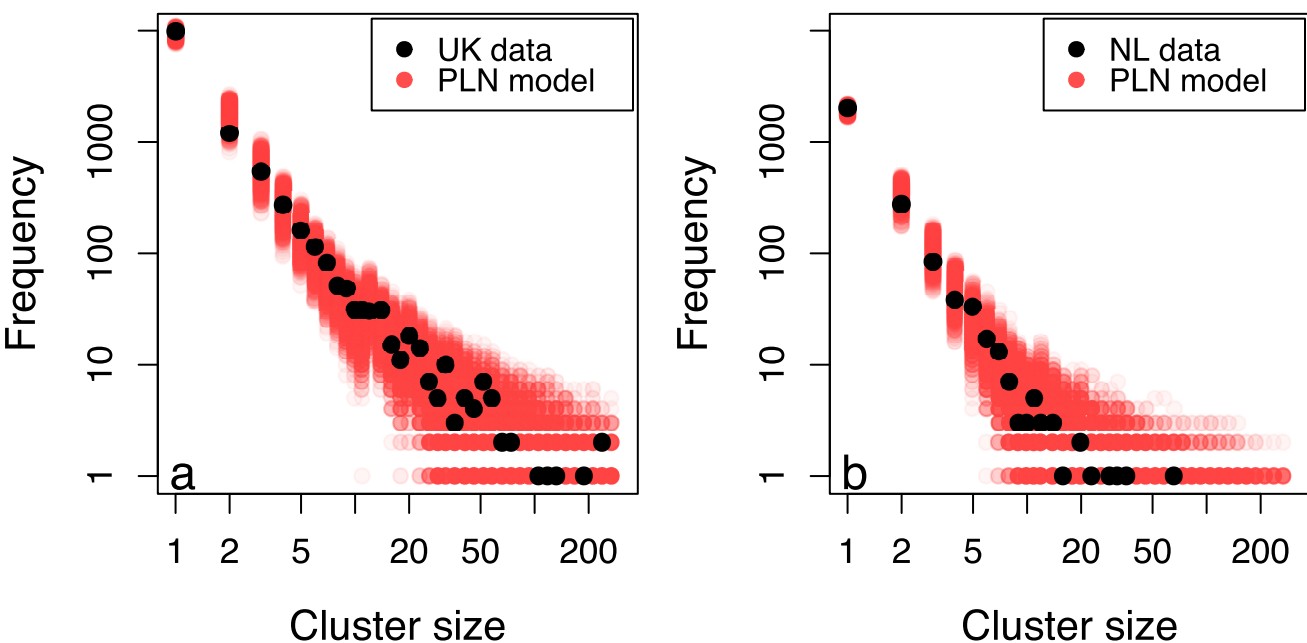

**Fig 3.** The distribution of cluster sizes for the UK (a) and the NL (b) with the distribution of cluster sizes produced by 1,000 iterations of the Poisson-lognormal model with parameters drawn from the posterior distributions.

transmission is not sustained (see section S1.4 and Figure H in S1 Text). Using Eq 2 (Methods), this means an average index case will generate 0.7 (0.4, 1.5) further cases. Furthermore, using the model, we find that clusters with more than 10 cases have an average reproduction number greater than 0.9.

In NL, the reproduction number using all data from 2004 to 2015 was 0.33 (0.22, 0.50) and since 2010 this reduced to 0.25 (0.14, 0.48). Even considering onward transmission chains, an average index case in the Netherlands generates 0.3 (0.16, 0.92) further cases.

### The role of superspreaders and onward transmission

Fig 4 illustrates the distribution of secondary cases by infectee reproduction number. From the UK data, we estimate that 84% (71%, 90%) of cases did not generate any secondary cases, therefore current control measures are adequately preventing onward transmission in the majority of cases. A further 10% (5%, 17%) of cases generated one secondary case only; they generated 25% (11%, 42%) of cases infected in the UK (Fig 4). 0.47% (0.1%, 0.9%) of cases generated more than 10 secondary cases, and could be considered "superspreaders". These superspreaders were responsible for 30% (4%, 62%) of secondary recently transmitted cases.

In NL between 2010 and 2015, 88% (77%, 96%) of cases did not generate any secondary cases and 8% (3%, 17%) of cases generated one secondary case, resulting in 34% (9%, 61%) of cases infected in NL. Superspreaders comprised 0.3% (0.02%, 0.8%) of all cases, and were responsible for 19% (1%, 52%) of recently transmitted cases.

By scaling down the transmission parameters in the UK, we estimate that if the UK were able to bring local transmission in line with the NL, they would be able to achieve a 17% (13%, 23%) reduction in incidence, equivalent to preventing 538 (266, 818) cases per year.

## Discussion

Tuberculosis (TB) remains a public health concern in low-incidence countries. As the majority of cases in low-incidence countries are foreign-born, impact of controlling recent transmission on overall TB burden is not clear.

Here, we presented methods for exploring and interpreting the full distribution of TB cluster sizes within a country in terms of recent transmission. Using data from the UK and NL, we find that the vast majority of cases did not transmit the infection. Less than 1% of cases caused more than 10 secondary cases and might be defined as "superspreaders". Overall, the average reproduction number is less than a half in both countries.

Superspreading, where a small proportion of cases generate a disproportionate number of secondary cases, is a common feature of many infectious disease epidemics[20]. Where superspreading dominates dynamics, targeted interventions perform better than population-wide measures, however identifying superspreaders can be challenging.

We estimate that onward transmission is substantially lower in the Netherlands than in the UK. Our estimate of the average reproduction number is consistent with previous estimates in low-incidence settings[21]. In particular, our estimate is in line with Borgdorff et al.'s estimates that also allow multiple introductions per cluster [22–24], suggesting that this is an important feature. Our estimate is lower than Ypma et al.'s estimate, which might be explained by the fact that they allow for the possibility of mutations within a cluster. In the UK, Vynnycky and Fine estimated that the effective reproduction number fell to well below one by 1990 [25]. We estimate that the reproduction number is now around 0.4 in the UK, and that reducing this to 0.25, in line with the Netherlands, could prevent one in six UK cases. A comparison of transmission, control measures and outcomes could elucidate the difference we observed between the UK and the NL. These would include the efficiency of contact tracing in the UK[26] and

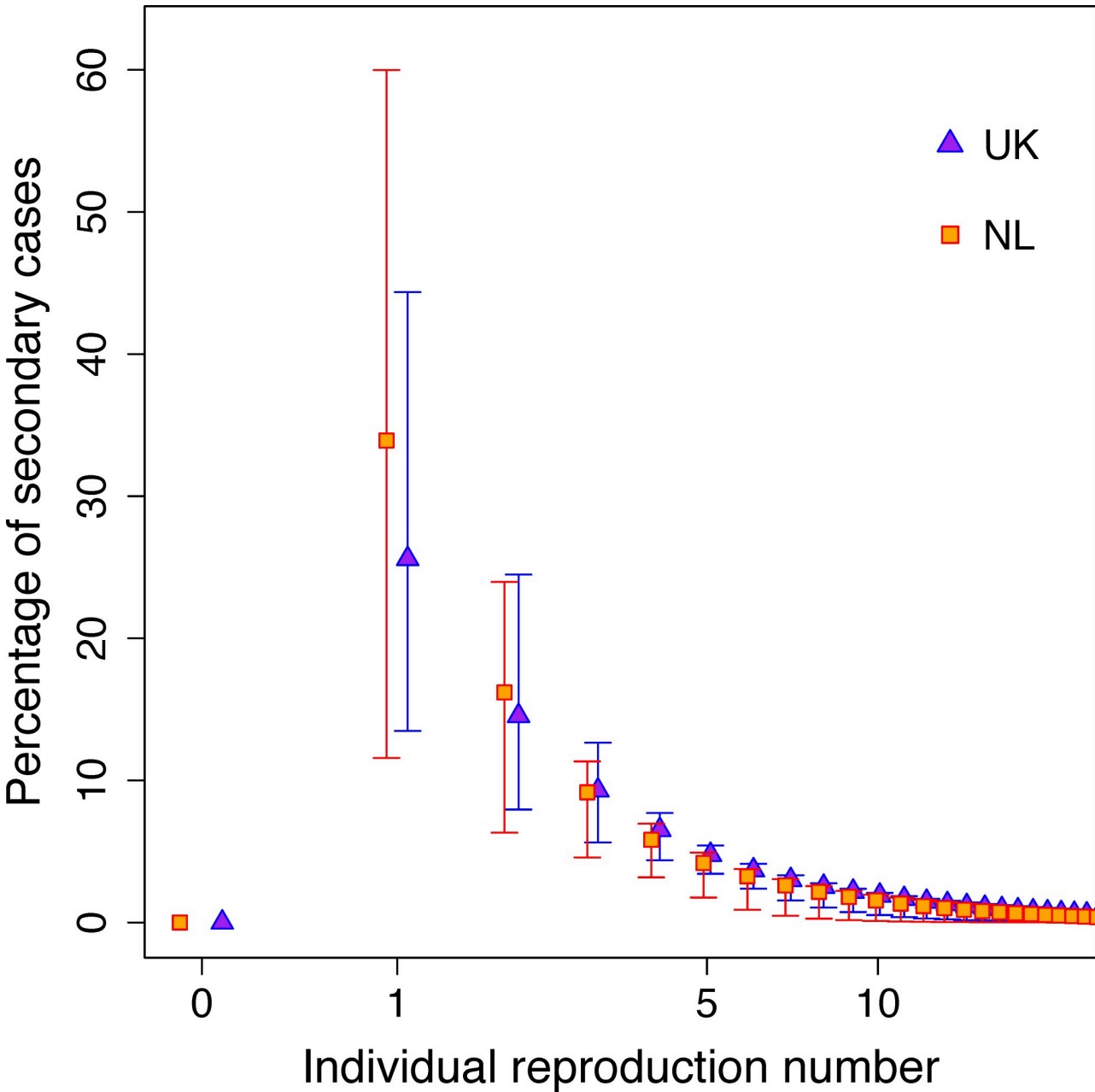

**Fig 4. The percentage of cases due to recent transmission against the reproduction number of the person who infected them.** The point estimates are the mean and the error bars are 95% credible intervals calculated using 10,000 parameter sets drawn from the posterior distribution of the model fit to the UK and NL data between 2010 and 2015.

the NL[27], household transmission[13], and different transmission rates between migrant groups[22].

Whole Genome Sequencing (WGS) is increasingly being used for genotyping in high-resource settings[28], having been introduced in 2018 in the NL and 2017 in the UK. In general, WGS analysis (using a 12 nucleotide difference threshold [29]) results in smaller clusters relative to MIRU-VNTR. However, because our method includes multiple independent importations per cluster the overall results are likely to be consistent between typing methods.

Applied to WGS data, our methods will provide an independent estimate of transmission, once the pipeline for TB DNA sequencing has been standardised across countries, consensus reached regarding the cut-off number of SNPs to be used for cluster definition[29] and multiple years of data have accumulated. With sufficient data, WGS can be used to re-construct transmission trees and directly estimate reproduction numbers[30]; however in many outbreaks WGS alone is not sufficient, and needs to be combined with epidemiological data and statistical inference[10].

The power of our method lies is the unification of clusters across multiple scales, and is therefore robust to missing data. However, the approach we used does have limitations. Firstly, the model did not include temporal or regional differences in transmission: these will be areas for future development. Further, we assumed that clusters were fully observed. In reality, only culture-confirmed cases can be genotyped and transmission within a cluster may be on-going. In 2015, genotyped cases represented 60.1% of all cases in the UK and 67% of all cases in the NL. The data are right-censored because clusters may not have run their full course; this will apply particularly to strains that appeared for the first time towards the end of the datasets. An inherent limitation of using a terminal branching process model is that we assume that transmission is not sustained without importation from outside the UK/NL or reactivation of old infections. The steady decline in incidence over the study periods suggests that this assumption is reasonable on average, although transmission is most likely sustained in the largest clusters. We did not capture the role of genetic mutation in generating new clusters, thereby potentially underestimating the contribution of recent transmission.

In summary, we observed consistent properties between TB clusters, irrespective of size, origin or country. We find that TB cluster sizes in low incidence countries can be captured by a simple model of importation and transmission. This work will contribute to a more well-developed understanding of TB transmission patterns in low incidence countries and how genotyping can be used for epidemiological inference. Control policies, such as contact tracing, aimed at limiting spread still have a role to play in eliminating TB in low-incidence countries.

## Methods

### Data sources

**UK data.**   The analysis was conducted using TB notifications collected through the Enhanced Tuberculosis Surveillance (ETS) system in England and Wales and the Enhanced Surveillance of Mycobacterial Infections (ESMI) system in Scotland. The following data for TB notifications were used: year of notification (2010 to 2015 inclusive), country of birth, disease type (pulmonary with or without extra-pulmonary or extra-pulmonary only), strain type (at least 23 out of 24 loci mycobacterial interspersed repetitive unit-variable-number tandem repeat (MIRU-VNTR) type), cluster name (assigned by a PHE naming tool based on strain type) and whether a case was categorised as clustered (yes/no).

**NL data.**   Data from the NL were extracted from the Netherlands Tuberculosis Register. MIRU-VNTR typing has been systematically conducted in the NL since 2004. As for the UK data, we extracted year of notification (2004–2015), country of birth, disease type (pulmonary or extra-pulmonary), strain type 24 loci MIRU-VNTR type.

**Defining clusters.**   Cluster size was defined as the number of cases with an indistinguishable MIRU-VNTR profile, where clusters of size 1 are cases with a unique 24 loci VNTR profile. Cases with a single missing locus that matched 23 loci of another cluster were considered part of that cluster[31]. Cluster sizes were binned logarithmically to retain the distribution shape while minimising noise due to low numbers of large clusters[32].

**Statistical model.** A feature of clusters size distributions in the UK and the NL, is that the proportion of clusters greater than a given size declines linearly with cluster size in log-log space. In order to characterise the distribution of clusters sizes across multiple scales, we fit a power law function of the form $P(x) \sim x^{-\alpha}$ to the cluster size distributions and assess the fit by calculating an associated p-value[33]. Two parameters are estimated: $x_{min}$, the minimum cluster size that is consistent with a power law, and $\alpha$, the exponent of the power law. The two parameters are estimated by minimising the Kolmogorov–Smirnov (KS) statistic, implemented in the poweRlaw R package[34]. 95% confidence intervals and a p-value are calculated. Within this framework, larger p-values indicate a better fit to the power law model than smaller values–see [33].

**Developing a mechanistic mathematical model.** Although the power law function estimated above provides a statistical description of the data, we were interested in finding a mechanistic explanation for the distribution of cluster sizes.

In order to do this, we used a mortal branching process model[18,35] with importation of infection to describe the process by which TB clusters are generated and evolve in low incidence settings. The central premise behind the model is that every diagnosed case must have been generated by one of two mechanisms, in a similar structure to household transmission models[36]: A) infection was acquired abroad or before the observation period (referred to as an imported or non-recent infection/reactivation) or B) the case was infected in the country during the observation period (interpreted as recently transmitted infection). Assuming these two mechanisms is broadly consistent with the data: in the UK, 81% of cases infected with a unique genotype were born outside the UK, compared to 70% of clustered cases.

For each unique genotype X, we assume that the first case cannot be due to a recent transmission event, i.e. it was either infected abroad or before routine genotyping. Each case $i$ generates $r_i$ secondary cases infected with genotype X where $r_i$ is drawn from a probability distribution. We did not differentiate pulmonary cases from extra-pulmonary cases, as there is no evidence of a correlation in pulmonary status between infector and infectee and a previous study of NL cluster sizes[18] found that including extra-pulmonary cases did not affect estimates of the reproduction number.

In addition to recently transmitted cases, we assume that for each case $i$ infected with genotype $X$, an additional, independent case also infected with genotype $X$ is diagnosed with probability $p$. This process is repeated for every case in the cluster, i.e. $C(X)$ times for a cluster with $C(X)$ cases. This results in a binomial distribution $Bin(C(X),p)$.

Each of the recent and non-recent cases have the opportunity to generate further secondary cases; this process is repeated until no new cases are generated. The branching process steps are as follows:

1. Start with the index case of a new cluster. Create a list of cases, $L$ containing a single case, $L = \{1\}$, such that the number of cases, $n = 1$;

2. For each case $i \in L$, draw the number of secondary cases produced by $i$, $r_i$, from the relevant distribution (Poisson lognormal or negative binomial) and add $r_i$ cases to the end of the case list, such that $n = n + r_i$;

3. Draw a random number between 0 and 1; if this is less than probability $p$, generate an imported case and add it to the end of the case list, such that $n = n + 1$;

In order to fit this model to cluster size data without further complexity, we impose the assumption that the average number of secondary cases per case must be greater than or equal to zero and less than one, $0 \leq E(r_i) < 1$, justified by the low and declining incidence in the two countries.

**Distribution of secondary cases per individual.** Previous analyses have modelled the number of secondary cases per TB case using a negative binomial distribution [18,20], which arises when the expected number of secondary cases per individual, $\lambda$, follows a Gamma distribution, $\lambda \sim \Gamma(k,\theta)$, with dispersion parameter $k$ and scaling parameter $\theta$. The average number of secondary cases per individual is given by $R = k\theta$.

We compare the negative binomial model for the distribution of within-country secondary cases with a Poisson-lognormal model. A Poisson-lognormal distribution is frequently used in ecological literature as an alternative to a negative binomial to describe species abundance for communities with many rare species[37]. It arises when the logarithm of the expected number of secondary cases per individual, $\log(\lambda)$, follows a normal distribution with mean $\mu$ and variance $\sigma$, $\lambda \sim \log N(\mu,\sigma)$. In a lognormal distribution, the average number of secondary cases per individual is given by

$$R = \exp(\mu + \sigma^2/2). \tag{Eq 1}$$

As $R<1$, the total number of additional cases due to an average imported case is calculated as the sum of a geometric series:

$$R/(1 - R). \tag{Eq 2}$$

We define a "superspreader" as a TB case in the above model that generates more than ten secondary cases. Using the model, we explore the impact of superspreaders by considering the proportion of secondary cases generated by persons with different reproduction numbers.

We use the model to estimate the impact of reducing transmission within the UK to match transmission within the Netherlands. We re-run the model with the estimated UK importation rate but scale the log mean of the Poisson lognormal distribution by a factor $\overline{\mu_{NL}}/\overline{\mu_{UK}}$, where $\overline{\mu_{NL}}$ is the average mean of the log normal distribution estimated for the NL and $\overline{\mu_{UK}}$ is the average mean of the log normal distribution estimated for the UK. The number of cases is totalled for the alternative scenario with lower transmission and compared to the total number of cases under the UK fitted model.

**Fraction of imported cases.** We estimate the proportion of cases due to recent transmission by recording the number of cases infected via direct transmission and the number of cases generated by importation during each simulation.

**Model fit.** In contrast to previous approaches that have used exact likelihood methods for fitting cluster size models to data[18,35], we use *Approximate Bayesian Computation (ABC)* [38,39]. In ABC, the likelihood is approximated by distance metrics based on summary statistics derived from the data and a realisation of the model, therefore can naturally incorporate the impact of sampling and importation. We use the Majoram MCMC search algorithm implemented in the R package EasyABC[40].

We estimated three model parameters: two for the distribution of secondary cases (either negative binomial or Poisson lognormal) and one for the importation rate. We assumed uniform prior distributions and imposed prior constraints that all parameters are greater than zero and that the reproduction number is greater than or equal to 0 and less than one. The target summary statistics were the number of observed clusters of a given size, logarithmically binned for a fixed number of bins. Using logarithmic binning attempts to compensate for the larger number of data points for lower cluster sizes. For $N$ bins, covering a range of cluster sizes from 1 to $C_{max}$, each bin is of length $\max(1, exp(n \log C_{max}/N))$ for $n = 1,\ldots,C_{max}$. We chose 50 bins, and the maximum cluster size was 300. For each set of proposal parameters, we simulated the model and binned the resulting cluster sizes in the same way as the data. The

distance between the model and the data was calculated using the Euclidean distance:

$$\sum_{i=1}^{nbins} \sqrt{\left(D_i - M_i\right)^2},$$

where $D_i$ is the number of observed clusters in the $i$th bin and $M_i$ is the number of clusters in the $i$th bin as predicted by the model.

We assessed model fit via two statistics: the proportion of clusters that are of size 1 (i.e. unmatched cases) and the number of cluster sizes that appear exactly once–see reference [17] for a further discussion of these quantities. Together, these two values capture characteristics of TB cluster size distributions across multiple settings with a high proportion of unmatched cases and larger clusters.

From the posterior distributions, we extracted the average number of secondary cases per individual ($R$), the degree of dispersion and the proportion of cases that are due to recent, within-country transmission. Unless otherwise stated, we report the mean from the posterior distribution and 95% credible intervals in brackets, calculated as the 2.5th and 97.5th quantiles of the posterior distributions.

## Supporting information

**S1 Text.** Details of the models: S1.1) Posterior distributions for the model parameters; S1.2) Comparison between the Poisson lognormal model and the negative binomial distribution model fits for the UK and the NL; S1.3) Posterior distribution for the proportion of cases not due to recent transmission; S1.4) Posterior distribution for the reproduction number in the UK and the NL.
(PDF)

**S1 Data. Number of clusters by size for the UK and the NL.**
(CSV)

## Acknowledgments

Thanks to Rolf Ypma for discussing his paper and early comparisons.

## Author Contributions

**Conceptualization:** Ellen Brooks-Pollock, Leon Danon, Andrew M. T. Pollock, Maeve K. Lalor.

**Data curation:** Hester Korthals Altes, Jennifer A. Davidson, Dick van Soolingen, Colin Campbell, Maeve K. Lalor.

**Formal analysis:** Ellen Brooks-Pollock.

**Funding acquisition:** Ellen Brooks-Pollock.

**Investigation:** Ellen Brooks-Pollock, Hester Korthals Altes, Jennifer A. Davidson, Maeve K. Lalor.

**Methodology:** Ellen Brooks-Pollock, Leon Danon, Hester Korthals Altes, Andrew M. T. Pollock.

**Software:** Leon Danon.

**Validation:** Ellen Brooks-Pollock, Leon Danon, Hester Korthals Altes, Dick van Soolingen.

**Visualization:** Ellen Brooks-Pollock.

**Writing – original draft:** Ellen Brooks-Pollock.

**Writing – review & editing:** Ellen Brooks-Pollock, Leon Danon, Hester Korthals Altes, Jennifer A. Davidson, Andrew M. T. Pollock, Dick van Soolingen, Colin Campbell, Maeve K. Lalor.

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
