## [Decision Letter · Decision Letter 0]

4 Sep 2019

Dear Dr Brooks-Pollock,

Thank you very much for submitting your manuscript 'A universal model of tuberculosis clustering in low incidence countries reveals more transmission in the United Kingdom than the Netherlands between 2010 and 2015' for review by PLOS Computational Biology. Your manuscript has been fully evaluated by the PLOS Computational Biology editorial team and in this case also by independent peer reviewers. The reviewers appreciated the attention to an important problem, but raised some substantial concerns about the manuscript as it currently stands. While your manuscript cannot be accepted in its present form, we are willing to consider a revised version in which the issues raised by the reviewers have been adequately addressed. We cannot, of course, promise publication at that time.

Sincerely,

Roger Dimitri Kouyos

Associate Editor

PLOS Computational Biology

Jason Papin

Editor-in-Chief

PLOS Computational Biology

[LINK]

Reviewer's Responses to Questions

**Comments to the Authors:**

Reviewer #1: De Brooks-Pollock and colleagues present an analysis of TB genotyping data in the United Kingdom and the Netherlands using branching processes. The methods are interesting and well suited to the objectives. The result that improvements in TB prevention in the UK similar to what has been done in the Netherlands could lead to an important decrease of TB incidence is convincing and important. However, several points of the paper need to be improved before publication, in particular the lack of rigor regarding the use and reporting of statistical methods.

Major points.

I have a problem with the way statistical inference and causality are handled in the results. First, the formulation suggests causality when only association is inferred (e.g. line 125: “having pulmonary disease increased the likelihood of belonging to a cluster”). In this example, the association could be explained by confounding factors, and even if causality exists it could go on the other direction, from belonging to a cluster to having pulmonary disease. The authors need to be a lot more careful with this very basic concept.

Second, the authors need to choose a framework for statistical inference and stick to it. If a null hypothesis significance testing framework is chosen (as suggests the use of p values), then the authors should systematically report effect sizes, confidence intervals and exact p-values for each performed inferred relationship, and not for instance R-squared values (line 127), nothing at all (line 135) or only “p value >0.10” (line 139).

Third, I’m very surprised to see the authors interpret the absence of statistical significance as an absence of effect, especially since this common mistake got a lot of publicity a few months ago with the Nature paper by Amrhein, Greenland and McShane. The authors should reformulate their conclusions on lines 126 (“had no effect”), 138 (“no consistent relationship”) and 345 (“there was no association”).

Fourth, a visual inspection of the fit (which is not even shown) is not sufficient evidence to infer that a power law “describes well” the data (line 143) or that a Poisson-lognormal model “captures the entire distribution” better than a negative binomial model. The authors should use model selection tools (AIC, DIC, cross-validation…) to be able to support such claims. A plot comparing the fits of different models might also help convince the readers.

Minor points

- Abstract/intro: The given intervals should be systematically defined.

- Line 87: I think the authors mean exactly the opposite.

- Line 103: The authors need to define what recent transmission means earlier.

- Lines 103-204: The authors should make it clear that these results are predictions conditional on the chosen model.

- Line 107: The title of the paragraph is “cluster size distribution” but no distribution is shown beyond the proportion of clusters of size 1. I would suggest showing the histograms in Fig. 1, together with the fits as suggested before.

- Line 115: No explanation is given regarding the time limits applied here.

- Line 184: Figure 3 is absolutely terrible and impossible to understand. The authors should definitely find a better way than pie charts to visualize the distribution of secondary cases.

- Line 262: Including cases that were not genotyped wouldn’t alter the estimates of transmission only if these cases are missing at random. This should be discussed in the context of the UK and the NL.

- Line 396: Where does this definition of superspreading come from? If it was used in previous works a reference should be added.

- Line 416: It is a bit clumsy to qualify approximate bayesian computation as an exact likelihood method.

Reviewer #2: This study analyses large data sets of clusters of tuberculosis based on MIRU-VNTR typing from the UK and from the Netherlands. In both places the incidence of TB is dropping so the effective reproduction number is below one. By fitting a model that includes the importation of new cases and a subcritical branching process to the data the authors are able to estimate the effective reproduction number and assess the predicted distribution of cluster sizes under the model. The focus is on pulmonary disease (the proportion of cases that are pulmonary is similar across countries and across cluster sizes). Fitting is performed with approximate Bayesian computation. The authors highlight the importance of superspreaders who transmit >10 cases each and discuss the epidemiological implications of reducing disease transmission.

This study makes excellent use of the large genetic data set; the model is simple yet able to reproduce features of the data successfully. A major issue, however, is that the manuscript feels incomplete at times - it's as if details are assumed to be obvious or already known or perhaps unimportant. Crucial details are missing particularly in the description of the simulation model and the ABC inference. I think it would be better to supply details especially since the current manuscript is not terribly long. Some examples are given below among specific comments and suggestions.

- Title: What is meant by "universal" in the title?

- Luciani et al 2008, Infection, Genetics and Evolution, considered the distribution of TB cluster sizes using population models. It could be worth looking at as part of the background information.

- L 251. I appreciate the argument that WGS alone may not provide enough information to estimate reproduction numbers, but this sentence is a bit unclear. I would explain why in terms of genetic markers and mutation rates.

- L 268 "However previous analysis found that right censoring..." This sentence is vague and cryptic -- spell out what the problem is. Right censoring of what? Why did it not affect the overall results? What is the meaning of "overall" results?

- L 290 This sentence mentions "at least 23 loci" before defining the 24-locus MIRU-VNTR typing. I would define VNTR first for clarity.

- L 354 m is the result of a binomial distribution; p is given but what is the "n" (number of trials) parameter? At first I thought it might be prevalence but I saw that later it is the quantity C(X), though I found that confusing as I'll explain below.

- L 359 "(see results)" Be more specific; say "see Figure 1" if that's what is meant.

- L 367 I find this equation unintuitive in the way it defines C(x) recursively. I can see that in a subcritical branching process the sum of r_i can go up to the total number of cases, but I don't understand the binomial term. How is the expression used computationally when the binomial term requires C(x) which it also contributes to? Could you please clarify and/or give details of how the expression is actually used in the simulation?

- L384 Use of the Poisson-lognormal. This is fair enough but provide a reference from the ecological literature, e.g. Bulmer 1974, Biometrics.

- L395 Eqn 2. Where does this come from? Explain or give a reference. I would also add nearby the condition that R<1.

- L 396 Is a superspreader one who generates more than 10 cases under the same model? Or has a different underlying rate? I believe it's the former but please clarify.

- L 415 The paragraph about the application of ABC is too brief in my opinion. I don't even know what parameters are being estimated. E.g. is p an unknown parameter to be estimated or set to some value? Give details of the prior distributions for them and the rationale for the choices. What's being estimated and what is fixed and assumed to be known?

What distance metrics are used? What do the posterior distributions of the parameters look like? Would be good to see them along with credible intervals, as they are the values that generate the model predictions in Figure 2.

I assume the "95% CIs" given in L170-204 are the credible intervals from the posteriors. If so, make that clearer. Since there are multiple ways to compute credible intervals, how are they actually computed here? "CI" is often read as "Confidence interval" which is a frequentist concept and presumably not what is meant here; clarify what you mean.

- Fig 1 The panel labels seem to be missing. On the pulmonary axis, perhaps add "UK" and "NL" somewhere.

- Fig 2. Missing panel labels again.

- Fig 3. The pie chart is not visually helpful. There is a large sector and many small sectors. The scrunched up sectors have to be expanded to show another unhelpful pie. Could you find another way to show the data? A histogram or a table perhaps?

**Have all data underlying the figures and results presented in the manuscript been provided?**

Reviewer #1: None

Reviewer #2: Yes

PLOS authors have the option to publish the peer review history of their article (what does this mean?). If published, this will include your full peer review and any attached files.

Reviewer #1: No

Reviewer #2: No

---

## [Decision Letter · Decision Letter 1]

16 Dec 2019

Dear Dr Brooks-Pollock,

Thank you very much for submitting your manuscript, 'A model of tuberculosis clustering in low incidence countries reveals more transmission in the United Kingdom than the Netherlands between 2010 and 2015', to PLOS Computational Biology. As with all papers submitted to the journal, yours was fully evaluated by the PLOS Computational Biology editorial team, and in this case, by independent peer reviewers. The reviewers appreciated the attention to an important topic but identified some aspects of the manuscript that should be improved.

We would therefore like to ask you to modify the manuscript according to the review recommendations before we can consider your manuscript for acceptance. Your revisions should address the specific points made by each reviewer and we encourage you to respond to particular issues Please note while forming your response, if your article is accepted, you may have the opportunity to make the peer review history publicly available. The record will include editor decision letters (with reviews) and your responses to reviewer comments. If eligible, we will contact you to opt in or out.raised.

- Supporting Information uploaded as separate files, titled 'Dataset', 'Figure', 'Table', 'Text', 'Protocol', 'Audio', or 'Video'.

We hope to receive your revised manuscript within the next 30 days. If you anticipate any delay in its return, we ask that you let us know the expected resubmission date by email at ploscompbiol@plos.org.

Sincerely,

Roger Dimitri Kouyos

Associate Editor

PLOS Computational Biology

Jason Papin

Editor-in-Chief

PLOS Computational Biology

[LINK]

Reviewer's Responses to Questions

**Comments to the Authors:**

Reviewer #1: I am satisfied with the revision.

Reviewer #2: The manuscript is improved and most of my suggestions/queries have been incorporate/addressed.

However, I would like to see the description of the model further developed near lines 354-365. The description of the branching process model is better but needs more work. The dummy variable i is used in a confusing way. First, it should be incremented somewhere - otherwise it stays at i=1 permanently and the loop can't be exited. Second, all cases become "relabelled" with respect to i after each "round" but this is not made explicit. Third, the value i=1 starts as the index case but after the first round i=1 is no longer the index case.

The equation on L364 still seems odd and misleading to me. As written, C(x) depends on itself but actually, this is a recursive function which is evaluated iteratively as the algorithm shows. This means that the C(x) on the left hand side is not actually the same quantity as the C(x) on the right hand side. After each round the C(x) must be updated.

I think the iterative structure of the calculations should be made more explicit. My understanding of the algorithm (and the accompanying equation) is as follows.

As described in the manuscript r_i is the number of secondary cases per case, distributed as specified by the model. Define C(X,j) to be the total number of cases of genotype X after j iterations and M(X,j) to be the number of imported cases (of genotype X) after j iterations.

1. Set initial conditions:

j = 0,

C(X,0) = 1 and r_i = 1 when j = 0 (the index case),

M(X,0) = 0 (no imports until the outbreak has begun)

2. Compute

M(X,j) ~ Binomial(C(X,j), p)

C(X,j+1) = \\sum_{i=1}^{C(X,j)} r_i + M(X,j)

3. Assign C(X,j) := C(X,j+1) and increment j to j+1

4. Repeat the recursion (steps 2 and 3) until C(X,j+1) = 0.

Minor comments.

L267 comma instead of full-stop/period in front of "thereby"

L276 reword by moving clause "such as contract tracing" next to "Control policies"

L420 Are all the priors uniform (or "flat")? Whatever they are, supply the information.

L421 I'm assuming that R>0 is also a condition so that 0<r<1.

</r<1.

**Have all data underlying the figures and results presented in the manuscript been provided?**

Reviewer #1: None

Reviewer #2: None

PLOS authors have the option to publish the peer review history of their article (what does this mean?). If published, this will include your full peer review and any attached files.

Reviewer #1: No

Reviewer #2: No

---

## [Decision Letter · Decision Letter 2]

16 Jan 2020

Dear Dr Brooks-Pollock,

We are pleased to inform you that your manuscript 'A model of tuberculosis clustering in low incidence countries reveals more transmission in the United Kingdom than the Netherlands between 2010 and 2015' has been provisionally accepted for publication in PLOS Computational Biology.

In the meantime, please log into Editorial Manager at https://www.editorialmanager.com/pcompbiol/, click the "Update My Information" link at the top of the page, and update your user information to ensure an efficient production and billing process.

One of the goals of PLOS is to make science accessible to educators and the public. PLOS staff issue occasional press releases and make early versions of PLOS Computational Biology articles available to science writers and journalists. PLOS staff also collaborate with Communication and Public Information Offices and would be happy to work with the relevant people at your institution or funding agency. If your institution or funding agency is interested in promoting your findings, please ask them to coordinate their releases with PLOS (contact ploscompbiol@plos.org).

Thank you again for supporting Open Access publishing. We look forward to publishing your paper in PLOS Computational Biology.

Sincerely,

Roger Dimitri Kouyos

Associate Editor

PLOS Computational Biology

Jason Papin

Editor-in-Chief

PLOS Computational Biology

Reviewer's Responses to Questions

**Comments to the Authors:**

Reviewer #2: The description of the model now makes a lot more sense.

This paper is a fine contribution which I look forward to seeing published.

**Have all data underlying the figures and results presented in the manuscript been provided?**

Reviewer #2: Yes

PLOS authors have the option to publish the peer review history of their article (what does this mean?). If published, this will include your full peer review and any attached files.

Reviewer #2: No

---

## [Editor Report · Acceptance letter]

17 Mar 2020

PCOMPBIOL-D-19-01185R2 

A model of tuberculosis clustering in low incidence countries reveals more transmission in the United Kingdom than the Netherlands between 2010 and 2015

Dear Dr Brooks-Pollock,

I am pleased to inform you that your manuscript has been formally accepted for publication in PLOS Computational Biology. Your manuscript is now with our production department and you will be notified of the publication date in due course.

With kind regards,

Laura Mallard
